# Large High-Efficiency Thermal Neutron Detectors Based on the Micromegas Technology

**Georgios Tsiledakis [1,\*], Alain Delbart [1], Daniel Desforge [1], Ioanis Giomataris [1], Thomas Papaevangelou [1], Richard Hall-Wilton [2], Carina Höglund [2,3], Linda Robinson [2], Susann Schmidt [2,3,4], Alain Menelle [5] and Michal Pomorski [6]**

[1] IRFU, CEA, Université Paris-Saclay, F-91191 Gif-sur-Yvette, France; Alain.DELBART@cea.fr (A.D.); daniel.desforge@cea.fr (D.D.); ioanis.giomataris@cern.ch (I.G.); thomas.papaevangelou@cea.fr (T.P.)

[2] ESS, European Spallation Source ERIC, PO Box 176, SE-22 100 Lund, Sweden; richard.hall-wilton@esss.se (R.H.-W.); carina.hoglund@esss.se (C.H.); Linda.Robinson@esss.se (L.R.); Susann.Schmidt@ionbond.com (S.S.)

[3] Thin Film Physics Division, Department of Physics, Chemistry and Biology (IFM), Linköping University, SE-581 83 Linköping, Sweden

[4] IHI Ionbond AG-Industriestraße 211, CH-4600 Olten, Switzerland

[5] Laboratoire Léon Brillouin, CEA, Université Paris-Saclay, F-91191 Gif-sur-Yvette, France; alain.menelle@cea.fr

[6] Laboratory for Integration of Systems and Technology, CEA, Université Paris-Saclay, F-91191 Gif-sur-Yvette, France; michal.pomorski@cea.fr

\* Correspondence: georgios.tsiledakis@cern.ch

**Abstract:** Due to the so-called $^3$He shortage crisis, many detection techniques for thermal neutrons are currently based on alternative converters. There are several possible ways of increasing the detection efficiency for thermal neutrons using the solid neutron-to-charge converters $^{10}$B or $^{10}$B$_4$C. Here, we present an investigation of the Micromegas technology. The micro-pattern gaseous detector Micromegas was developed in the past years at Saclay and is now used in a wide variety of neutron experiments due to its combination of high accuracy, high rate capability, excellent timing properties, and robustness. A large high-efficiency Micromegas-based neutron detector is proposed for thermal neutron detection, containing several layers of $^{10}$B$_4$C coatings that are mounted inside the gas volume. The principle and the fabrication of a single detector unit prototype with overall dimension of ~15 × 15 cm$^2$ and its possibility to modify the number of $^{10}$B$_4$C neutron converter layers are described. We also report results from measurements that are verified by simulations, demonstrating that typically five $^{10}$B$_4$C layers of 1–2 μm thickness would lead to a detection efficiency of 20% for thermal neutrons and a spatial resolution of sub-mm. The high potential of this novel technique is given by the design being easily adapted to large sizes by constructing a mosaic of several such detector units, resulting in a large area coverage and high detection efficiencies. An alternative way of achieving this is to use a multi-layered Micromegas that is equipped with two-side $^{10}$B$_4$C-coated gas electron multiplier (GEM)-type meshes, resulting in a robust and large surface detector. Another innovative and very promising concept for cost-effective, high-efficiency, large-scale neutron detectors is by stacking $^{10}$B$_4$C-coated microbulk Micromegas. A prototype was designed and built, and the tests so far look very encouraging.

**Keywords:** gaseous detectors; micro-pattern gaseous detectors (MPGD; GEM; Micromegas; Microbulk; etc.); neutron detectors (thermal neutrons); detector modeling and simulations; boron carbide coatings

## 1. Introduction

Currently, neutron detectors with good performance are urgently needed to take full benefit of the high-intensity neutron beams produced by sources like the two large-scale neutron facilities, SNS (Spallation Neutron Source) in the United States (US) and J-PARC (Japan Proton Accelerator Research Complex) in Japan, as well as the future European Spallation Source (ESS) which will produce its first neutrons in 2022. However, in 2008, a big constraint came from the fact that the volume of $^3$He available was by far insufficient to cope with the demands for large-area detectors, resulting in a considerable cost increase of this gas. Due to the so-called $^3$He shortage crisis at that time, a coordinated initiative was launched by several neutron science institutes, and a roadmap was defined to support the development of position sensitive detectors using neutron converters other than $^3$He [1]. $^{10}$B-based detectors were demonstrated in the past years as efficient, cost-effective, and environmentally safe alternatives to $^3$He-based neutron detectors [2]. $^{10}$B-containing converter layers, in the form of $^{10}$B$_4$C coatings on metal substrates, that are mounted inside gas detectors are the subject of the investigations reported in this paper. However, the main disadvantage of such a set-up is the low achievable efficiency (~5% for thermal neutrons) when using a single conversion layer. The detection mechanism involves an atom of $^{10}$B capturing a neutron, thereby producing two ionizing particles, $^7$Li and $^4$He, which are emitted in opposite directions via the following reactions:

$$n + {}^{10}\text{B} \rightarrow {}^7\text{Li} \, (0.84 \text{ MeV}/\text{r} = 1.9 \; \mu) + {}^4\text{He} \, (1.47 \text{ MeV}/\text{r} = 3.6 \; \mu) + \gamma \, (0.48 \text{ MeV}) \, (94\%),$$

$$\rightarrow {}^7\text{Li} \, (1.02 \text{ MeV}/\text{r} = 2.2 \; \mu) + {}^4\text{He} \, (1.78 \text{ MeV}/ \text{ r} = 4.4 \; \mu) \, (6\%)$$

where energies and ranges in the converter film are indicated for each particle [3–7]. Since atomic boron is not seen as chemically stable enough and would be much harder to deposit as a high-quality coating, the chosen converter material is $^{10}$B$_4$C (boron carbide). $^{10}$B$_4$C is commercially available with a $^{10}$B enrichment of up to more than 97%. Compared to multi-wire proportional chambers (MWPCs), micro-pattern gas detectors (MPGDs) that are widely used in high-energy physics (HEP) to detect minimum ionizing particles (MIPs) offer better spatial resolution, counting rate capability, and radiation hardness; their fabrication is also more reproducible. Provided similar advantages are applicable to detect neutrons, MPGDs might contribute significantly to the development of neutron scientific instrumentation [8]. The Micromegas detector [9] is a very asymmetric double-structure detector. This detector is unlike its predecessors in that the two well-distinguished regions are no longer separated by a plane of wires but by a micromesh which separates the conversion space, the dimension of which typically ranges from 2 mm to 10 mm, from a small amplification gap that can be as small as 50 μm. Applying appropriate potentials to the electrodes, a very high electric field in the amplification region can be obtained with a low electric field in the drift region (Ed). In the conversion region, primary electrons are produced and are transported into the amplification region where the multiplication avalanche is created. The ratio between the electric field in the amplification gap and that in the conversion gap can be tuned to large values, as required for an optimal operation of the device. Furthermore, such a high ratio is also required in order to catch the ions in the small amplification gap: under the action of the high electric field, the ion cloud is quickly collected on the micromesh and only a small part of it, inversely proportional to the Ea/Ed ratio, escapes toward the conversion region. The n-TOF project (neutron time-of-flight facility) which measures neutron interaction cross-sections and beam sizes uses Micromegas equipped with B and $^{10}$B$_4$C converter films. Compared with solid-state technology like boron- or LiF-coated microstructured Si thermal neutron detectors, Micromegas has the advantage that it is relatively easy to make large surfaces with low cost. Moreover, it is a radiation hard detector that can operate in very high particle fluxes, has fast signals (~50 ns duration), and it can be insensitive to gammas. An innovative concept for a cost-effective, high-efficiency, large-scale neutron detector which associates a compact stack of multi-stage $^{10}$B$_4$C-covered meshes, with a Micromegas gaseous amplification was in development since February 2012, between the The French Alternative Energies and Atomic Energy Commission

(CEA)/Irfu institute (the inventor of Micromegas in 1996) and Laboratoire Léon Brillouin (LLB) (a major research infrastructure devoted to the study of condensed matter by thermal neutron scattering). The coating was deposited via direct current (DC) magnetron sputtering in an industrially scaled deposition system, either at the Linköping University or in the ESS detector coatings workshop, Linköping, Sweden, using deposition parameters similar to the ones presented in References [3–5]. This project was funded by the European Union's collaborative Seventh Framework Program for research, technological development, and demonstration under the NMI3-II Grant number 283883 (FP7-NMI3), being part of the contributions submitted to ESS [10]. The large high-efficiency multi-layered Micromegas thermal neutron detector, built under this framework FP7-NMI3, was presented in detail in Reference [11] and its main aspects are shown in the next section. Based on the same concept, CEA designed and simulated another neutron-sensitive Micromegas detector using the Microbulk Micromegas detector technology for the SINE2020 project that received funding from the European Union's Horizon 2020 research and innovation program under grant agreement No. 654000, materialized in ESS. The European Spallation Source (ESS) now under construction in Sweden, will produce intense beams of neutrons for determining the structure and dynamics of materials. Such intense beams present challenges for the current technologies used to detect these neutrons. The need for high-rate, high-resolution detectors for reflectometry applications, which are able to cope with requirements of the ESS, are common to all neutron and muon sources; therefore, this proposal is of wide synergetic benefit across the European neutron and muon facilities.

## 2. Simulations to Evaluate Electric Fields and Thermal Neutron Detection Efficiencies

A series of simulations were developed to model the structure of the detector and understand the parameters required to optimize the detector design for maximum neutron detection efficiency. The required electric field applied on the detector and the electron collection efficiency for different gas mixtures were evaluated by Garfield, Magboltz and neBEM (nearly exact Boundary Element Method) simulations. A single two-mesh detector unit is sketched in Figure 1. It consists of an aluminum box of 0.1 mm wall thickness, two nickel meshes that each are 4 $\mu$m thick, and three gas layers of 1 mm thickness each. The two meshes are two-side-coated with $^{10}B_4C$ and the aluminum entrance plate is one-side-coated on the inner side (see Figure 1). The electrons that are produced due to the neutron capture by a $^{10}B$ atom start drifting toward the anode by passing through the two nickel meshes. A multiplication factor F needs to be applied on the potential for each of the layers to magnify the collection efficiency of the electrons. Attributing a potential V0 at the anode and an electric field $E = -10$ V/mm, and if $(n - 1)$ is the number of nickel meshes, we end up with the formula $V(n) = V(n - 1) + F^{(n-1)}E \times D$. COMSOL™ [12] simulations obviously demonstrate that, by increasing the factor F (from 1 to 10), more field lines pass through the holes toward the anode, leading to better collection efficiencies for the produced electrons in the gas. For our calculations in this study, a multiplication factor of F = 5 was chosen as the most suitable and moderate approach with low potential values. Therefore, for a single two-mesh detector unit, the nominal applied potentials for F = 5 are [V0, V1, V2, V3] = −[300, 550, 600, 610] V.

Three-dimensional (3D) electrostatic field numerical simulations were carried out using the Garfield [13] simulation framework with the addition of the neBEM [14,15] toolkit in order to investigate how the thickness of the mesh affects the electron transparency. The outcome indicates the need of very thin meshes to achieve high electron transmission. Therefore, nickel meshes of 5 $\mu$m thickness were ordered, coated, and tested.

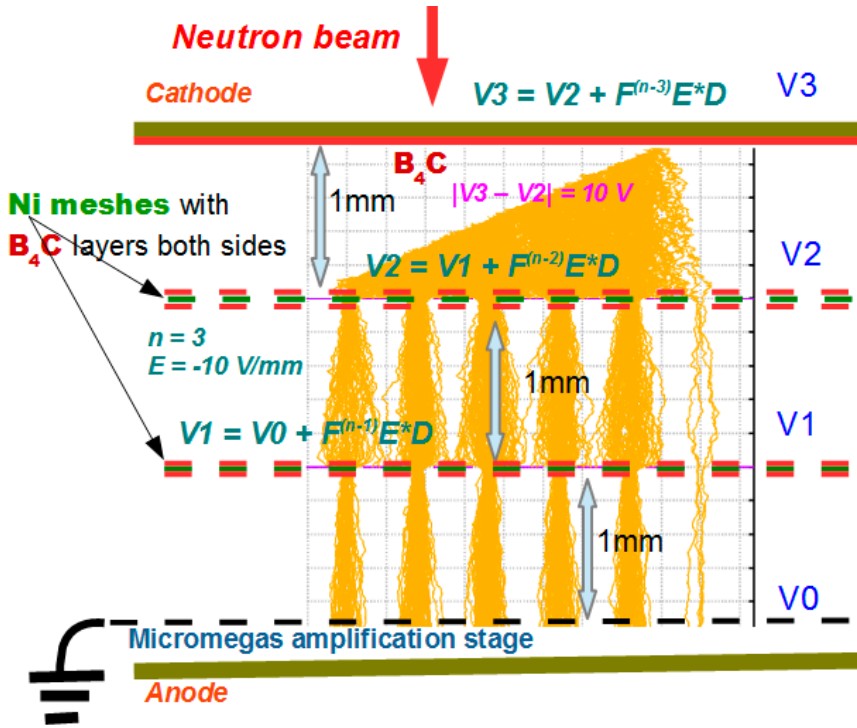

**Figure 1.** A single two-mesh detector unit with five layers of $^{10}B_4C$ (two two-side-coated woven nickel meshes and one one-side coated aluminum) and the electron transmission in a gas mixture of $CF_4$ obtained with Garfield simulations [11].

A comprehensive and accurate set of Monte Carlo simulations was performed to optimize the thickness of the $^{10}B_4C$ layer for maximizing the conversion efficiency, defined as the probability for a neutron to produce an $\alpha$ particle and/or a Li ion that enter into the gas gap. For the current work, the FLUKA Monte-Carlo simulation package [16,17] was used (simulation results using the GEANT4 platform were compared to cross-validate the code), where the low energy neutron transport package was activated and employed for thermal neutron detection. The demanded detection efficiency is given by the number of entries in the energy deposition histograms inside the gas volume divided by the total number of thermal neutrons hitting the detector. A $^{10}B$ layer thicker than 2–3 µm is not effective due to the absorption of the reaction products. In this case, the maximum efficiency that can be achieved with one single $^{10}B$ layer is ~4–5%. The multi-layer approach was examined in order to boost the thermal neutron detection efficiency. A tower of detector–converter layers could be a solution, but with the drawback of introducing many detectors and lots of material. The alternative approach is to build a tower of converter layers for each detector, having $^{10}B_4C$ deposited on thin metallic meshes placed in the drift region, resulting in less electronics and material. The applied electric field should be properly configured in order to drift the produced charges to the detector through the holes in the meshes. One module consists of a double-face Micromegas facing 7 + 7 $^{10}B_4C$ layers with a total thickness of ~1 cm (PCB: 0.2–0.3 µm, Ni 6 × 5 µm, 2 × micromesh, Al case 2 × 1 mm, where PCB is the anode printed circuit board). A single two-mesh detector unit with five layers of 2–3-µm-thick $^{10}B_4C$ coatings that is irradiated with a perpendicular point-like thermal neutron beam can record efficiencies of ~20%, as illustrated in Figure 2. In accordance with the simulations, a baseline design of a detection unit can be made of a 20-mm-thick stack consisting of seven $^{10}B_4C$ layers that are deposited on two sides of three meshes and on the inner side of the entrance window. Thermal neutron detection efficiency of such a unit is simulated to be 25%, but goes up to 39% with two units assembled back to back, and reaches 45% with a 45° neutron beam incidence. By using two double two-mesh or three-mesh detector units, neutron detection efficiencies of 49% and 57%, respectively, are achievable. With a 45° tilt of the full detector with respect to the neutron beam, efficiencies are as

high as 57% (assuming a two-mesh structure, with 20 $^{10}B_4C$ layers) and 64% (assuming a three-mesh structure, with 28 $^{10}B_4C$ layers) are possible. More details can be found in References [1,18].

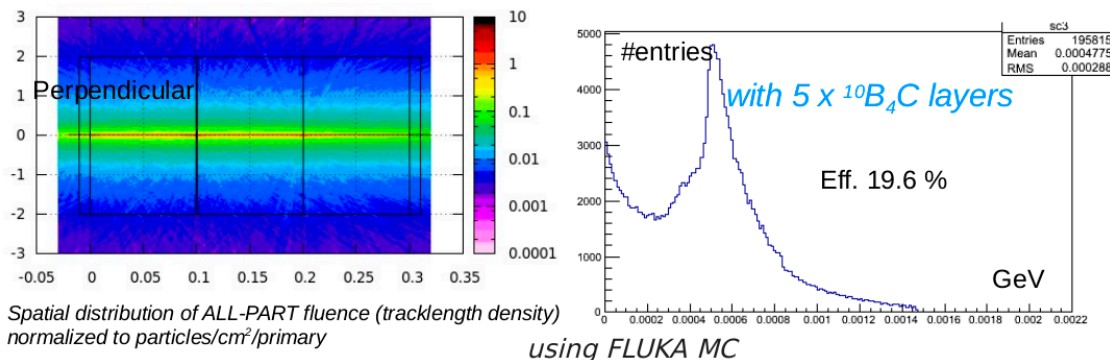

**Figure 2.** Thermal neutron detection efficiencies of a two-mesh detector unit (five layers of $^{10}B_4C$ with a thickness of 1.5 μm each) [18].

## 3. A Prototype Design: Construction and Tests

A prototype was designed and built to test the concept. It consists of a modular $15 \times 15 \times 2$ cm$^3$ gas chamber in which up to four meshes can be stacked and polarized above a Micromegas amplification structure, either a standard bulk Micromegas [19] or a Kapton Microbulk Micromegas [20]. The design can be seen in Figure 3. Its frame is $7 \times 7$ cm$^2$ and its active zone is $5.4 \times 5.4$ cm$^2$. The detector concept requires high-quality $^{10}B_4C$ coatings to be deposited onto different types of thin meshes (nickel, stainless steel, or copper-cladded Kapton foils) to enable an efficient neutron conversion process and a charge transport through the multi-layered structure onto the Micromegas amplification stage. A series of $^{10}B_4C$ depositions were done at Linköping University and in the ESS detector coatings workshop in Linköping, Sweden. Nickel meshes that were 5 μm thick and either 10 or 20% optically transparent were covered on both sides with 1.5-μm-thick $^{10}B_4C$ coatings. One of these meshes, 10% optically transparent, was assembled in the prototype above a bulk Micromegas amplification stage to perform a series of measurements with a $^{252}Cf$ neutron source.

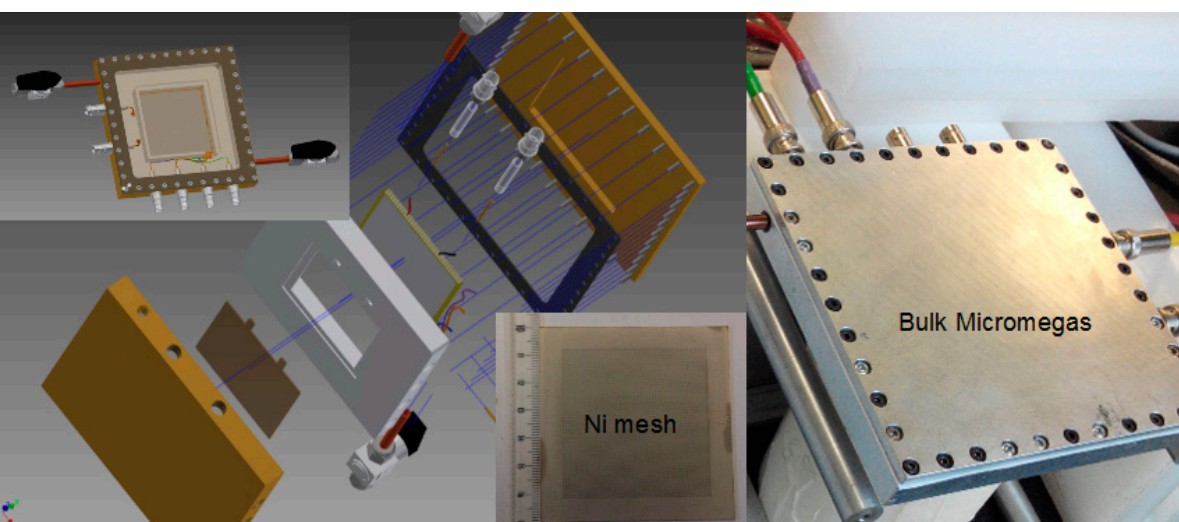

**Figure 3.** Design of the prototype detector unit and the experimental set-up at CEA/Irfu [11].

Our first set-up is shown in Figure 3. Here, the mesh channel is connected to ground, the drift is connected through a filter to a CAEN High Voltage (HV) power supply, and the anode is connected through a preamplifier (PA) to a CAEN HV as well. The $^{252}Cf$ neutron source was placed in the middle of a polyethylene cylinder. Different field configurations for various gas mixtures were tested and the

detector signal spectrum was measured and well reproduced by the simulations. The detector score counts from thermal neutrons originated from the $^{252}$Cf neutron source. The detector signal spectrum, as can be seen in Figure 4, was measured using a multi-channel analyzer program (MCA) to process detector signals produced by the detector, to measure the pulse height, and to obtain the pulse height spectrum, which is the number of counts as a function of the MCA channel.

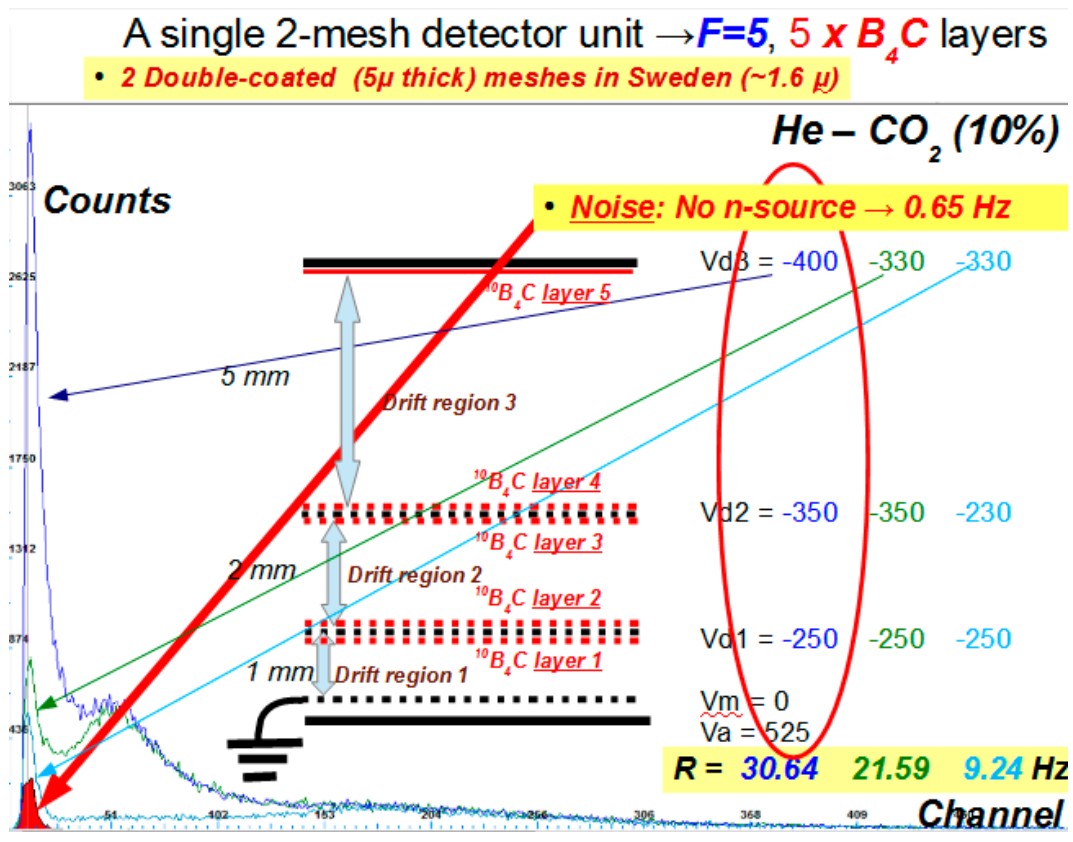

**Figure 4.** Data measured during 3000 s for each run, using a single two-mesh detector unit equipped with five layers of $^{10}$B$_4$C [11].

The next stage of this project was to build a two-mesh detector unit, having added two Ni meshes that were two-side coated with $^{10}$B$_4$C (1.6 μm thick) and a one-side coated aluminum end plate, and to measure and quantify the contributions on the counting rate from each of the five $^{10}$B$_4$C layers. In Figure 4, the experimental set-up has drift region-2 twice the length of drift region-1 in order to double the ionization leading to a higher electron transmission, while the thickness of drift region-3 is 5 mm for the same reason. The data indicate that the detector response is ~0.65 Hz (noise) when no neutron source is used. Having placed the detector on top of the neutron source and varying the applied voltages, the contribution of each $^{10}$B$_4$C layer to the counting rate can be estimated. Hence, a counting rate of R = 30.64 Hz was measured with contributions from all five $^{10}$B$_4$C layers where the applied drift voltages were [Vd1, Vd2, Vd3] = [−250, −350, −400] V, corresponding to a field multiplication factor F = 5. By inverting the Vd3 setting [Vd1, Vd2, Vd3] = [−250, −350, −330] V, $^{10}$B$_4$C layers 1–3 contribute, resulting on a measured rate of ~21.59 Hz, while, by setting [Vd1, Vd2, Vd3] = [−250, −230, −330] V, a rate of ~9.24 Hz was recorded, which comes from the $^{10}$B$_4$C layer 1. Higher recorded counting rates of ~36 Hz can be obtained with a higher multiplication factor of F = 10. We establish here that the proposed detector works; however, for obtaining more reliable results, we should optimize the applied voltages and the drift distances in order to increase the ionization, and maximize the electron transmission through the meshes by having the peaks of the rate distributions from different drift regions overlapping with each other. The performance of this configuration and set-up

was compared to a calibrated reference $^3He$ tube detector (see Figure 4). Measuring the thermal neutron efficiencies with both detectors we have [Signal − Background] $^3$He/[Signal − Background]Det. = 5.6. As a result, attributing an efficiency of ~100% in the $^3$He detector, the single two-mesh detector unit should have a thermal neutron detection efficiency of ~18% which was verified and validated with simulations (~19.6% expected from FLUKA MC).

An important issue that was also investigated in Reference [11] was the possible contamination of the neutron-induced signal from gammas and electronic noise using a gamma source $^{60}$Co. The results showed that the expected contribution is very low and could be erased by setting an energy threshold as low as ~50 keV. Thus, the proposed detector, depending of the applied voltages, can be completely insensitive to gammas [21].

This multi-layered bulk Micromegas detector prototype works fine and its performance agrees with the results anticipated from the modeling accomplished in NMI3. Nevertheless, the Ni meshes used in the bulk Micromegas detector are very thin and, thus, fragile. Furthermore, due to the thickness of the embodied PCB layer which supports the readout electrodes, the stacking of the detectors to enhance the thermal neutron detection efficiency is limited. Thick and more robust meshes (>20 μm) have very low electron transmission, while thin meshes (<5 μm) are easily deformed, making it potentially difficult to operate a large detection system with more than three layers per unit.

## 4. Using the Microbulk Technology

According to the "bulk" method, the amplification region of the Micromegas detector is produced as a single piece. A woven mesh is laminated on a PCB (anode printed circuit board) covered by a photoimageable polyimide film, and the pillars are made using a photochemical technique with insulation through the grid. The new "Microbulk" Micromegas manufacturing technique [22–25], based on Kapton thin-foil etching technology which was developed recently, can be used to fabricate detectors, and reading out anode and cathode strips in a single integrated structure. Micromegas detectors constructed with Microbulk technology have several advantages such as low detector material, high radiopurity, and better energy resolution. Moreover, the minimal amount of the material, and the small neutron sensitivity make these detectors appropriate for neutron beam experiments, such as nTOF, where Microbulks are being used for beam monitoring. The main advantage of the Microbulk detector is that there is no PCB and the readout pads are supported directly on 50-μm pillars which support the micromesh. Neutron scattering from such an arrangement should be very low and, hence, it should be possible to stack several layers behind each other without adversely affecting the incoming neutrons. $^{10}$B$_4$C can be deposited not only in the cathode, but also on the Microbulk surface, since it is a Kapton mesh coated with Cu, thus doubling the thermal neutron detection efficiency. Since one back-to-back microbulk detector unit can be very thin (~5 mm), a mosaic of such detectors could be easily built without limitation, using only three voltages (same cathode, mesh, anode voltages) avoiding electron transmission problems, with better spatial resolution and low material budget. One such unit could record thermal neutrons with efficiencies of ~17%. Simulations, based on FLUKA MC, show that, by placing four back-to-back Microbulk Micromegas detector units, the thermal neutron detection efficiency is 40% and, with a 45° tilt of the full detector with respect to the neutron beam, it can reach up to 45% (16 × $^{10}$B$_4$C layers that are 2 μm thick) [11]. A prototype was designed and built, being a modular [15 × 15 × 2] cm$^3$ chamber in which up to four Kapton Microbulk meshes can be stacked. Attempts were made to coat the Microbulk (Kapton) material directly with $^{10}$B$_4$C. Tests are ongoing with several samples of thicker Kapton meshes with different frame materials. A coating of a nickel plated microbulk with ~0.4-μm $^{10}$B$_4$C was achieved at CEA using a PVD (physical vapor deposition) sputtering machine. Measurements were performed with a coated Microbulk detector installed into the prototype unit placed on top of a polyethylene cylinder that has a $^{252}$Cf neutron source in its middle. Here, the anode channel was connected to ground, the drift channel was connected through a filter to a CAEN HV power supply, and the mesh was connected through a preamplifier (PA) to a CAEN HV as well. The detector score counts from thermal neutrons

originated from the $^{252}$Cf neutron source. The detector signal spectrum, as can be seen in Figure 5, was measured using a multi-channel analyzer program (MCA). The results look very encouraging and are in agreement with an independent pulse shape analysis. Thus, this coated Microbulk can record a thermal neutron detection efficiency of ~2.4 Hz. If the thickness of the $^{10}$B$_4$C foil is ~1.5–2 μm, an efficiency of ~9–10 Hz is expected.

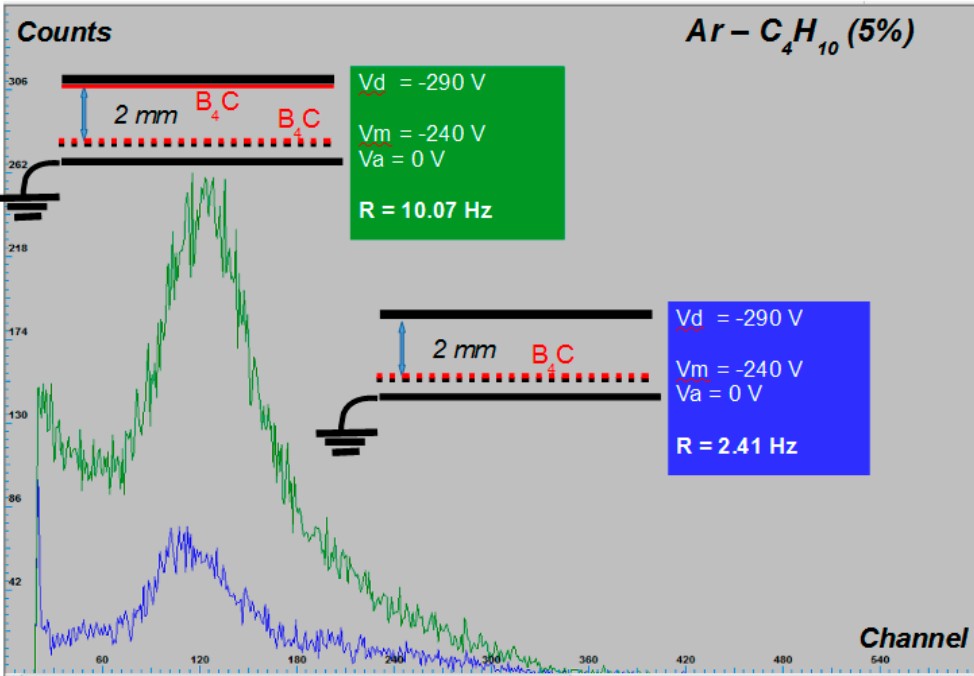

**Figure 5.** Data measured during 3000 s for two runs with one-side coated aluminum end plate and without coated end plate, using a Microbulk Micromegas coated with 0.4-μm $^{10}$B$_4$C and installed into a single detector prototype unit equipped with a $^{10}$B$_4$C one-side coated aluminum end plate (1.5 μm thick).

In the meantime, a simplified concept as an alternative to Microbulk stack is being developed, where the mesh of a bulk Micromegas detector is replaced by a Microbulk layer. Thus, a $^{10}$B$_4$C-coated mesh used in the multi-layered structure which was successfully built under the FP7-NMI3 program [11] is replaced by a gas electron multiplier (GEM) [26,27] foil. GEMs consist of a thin metal-clad insulating polymer foil into which conical holes are etched. Upon application of a moderate voltage to the metal layers, electrons, drifting into the holes, are amplified efficiently. The GEM foils used in our simulations and measurements consist of 12-μm-thick Kapton, two-side coated with 3-μm-thick copper, covered by a 1.5-μm-thick layer of nickel. Preliminary tests of the prototype were done, where the mesh was replaced by a Microbulk layer (GEM foil). This GEM foil mesh divides the drift region into two 2-mm regions. Carrying out 3D electrostatic field simulations (Garfield/neBEM) and comparing meshes of GEM type and metallic ones of the same total thickness leads to a conclusion that a high electron transmission should be expected using GEM-type meshes [11]. Measurements were performed with the same prototype detector unit placed on top of a polyethylene cylinder that has a $^{252}$Cf neutron source in its middle, having as a mesh a Microbulk GEM-type foil. Measurements show that a Kapton GEM-type mesh provides much better electron transmission than a thin metallic one, and is also cheap and robust, allowing it to be used for big surfaces (1 × 0.5 m$^2$). Recently, a one-side coating of a GEM foil with ~0.4-μm $^{10}$B$_4$C was achieved at CEA using a PVD sputtering machine. The detector signal spectra, as can be seen in Figure 6, were measured using a multi-channel analyzer program (MCA). The results look very promising, and the recorded thermal neutron efficiencies are

~3.1 Hz from the one-side coated GEM foil, and ~10.5 Hz when the coated GEM is combined with a $^{10}B_4C$ one-side coated aluminum end plate (~1.5 µm thick).

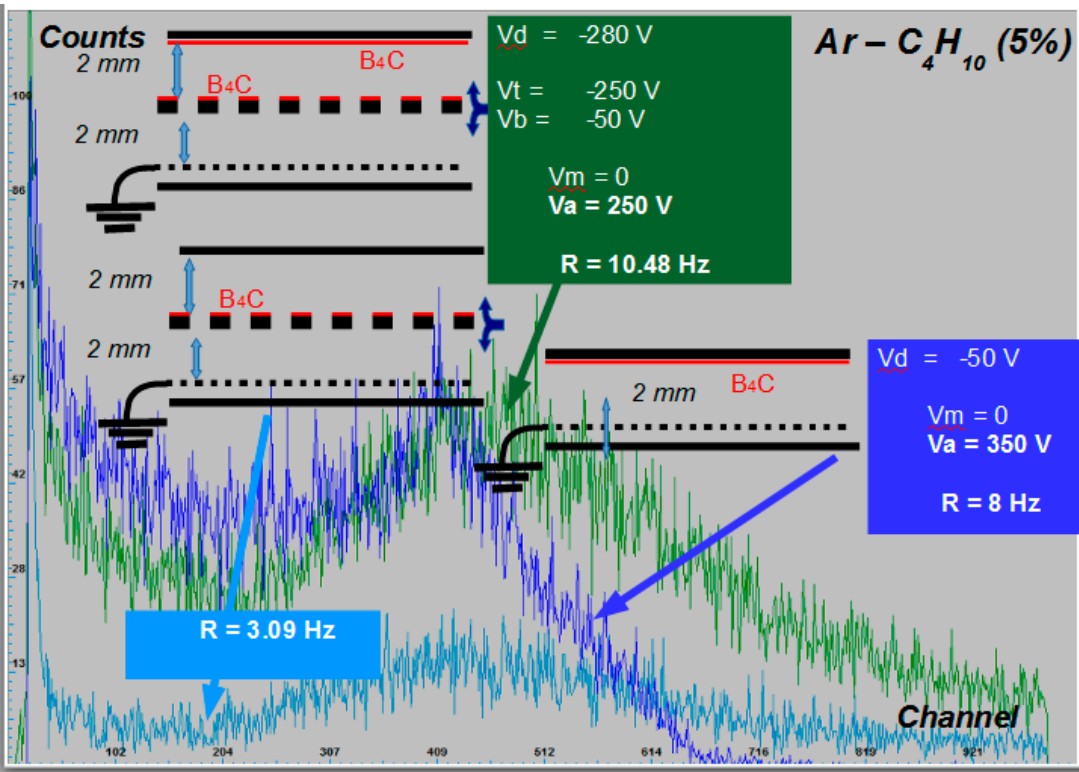

**Figure 6.** Data measured during 3000 s for each run, using a gas electron multiplier (GEM)-type Kapton mesh coated with 0.4-µm $^{10}B_4C$ and for three different configurations.

## 5. Conclusions

Several possible ways of building a cost-effective, high-efficiency, large-scale neutron detector, using the Micromegas technology and $^{10}B_4C$ neutron converter layers, were investigated. Our measurements illustrate that a single bulk Micromegas two-mesh detector unit, equipped with two metallic (Ni) thin meshes coated with $^{10}B_4C$ on both sides can record efficiencies of ~18%, as expected from simulations. The desired thermal neutron detection efficiency of ~50% can be achieved by placing two double two-mesh detector units without tilting the detectors. Nevertheless, the very thin Ni meshes that were used in order to achieve high electron transmission are fragile and could easily be deformed, preventing the application of very high voltages and, thus, limiting their use for large surface detectors. The aforementioned obstacle can be overcome by implementing the new Microbulk Micromegas technology that results in detectors with low material budget, low intrinsic radioactivity, and stability. A prototype was designed and built, and the tests with a single Microbulk Micromegas detector unit coated with $^{10}B_4C$ look very encouraging. A stack of such microbulk units coated with $^{10}B_4C$ is planned to be constructed. Another additional innovative and very promising concept is to use a bulk Micromegas equipped with GEM-type meshes instead of the fragile Ni meshes, coated with $^{10}B_4C$ on both sides, and resulting in a robust and large surface detector. Tests are ongoing to deposit $^{10}B_4C$ on such meshes.

**Author Contributions:** All authors contribute as a team to this published work.

**Funding:** This project received funding from the European Union's collaborative Seventh Framework Program for research, technological development, and demonstration under the NMI3-II Grant No. 283883 (FP7-NMI3), being part of the contributions submitted to ESS and from the European Union's Horizon 2020 research and innovation program under the grant agreement No. 654000, materialized in ESS (SINE2020). Richard Hall-Wilton, Carina Höglund, Linda Robinson, and Susann Schmidt would like to acknowledge the financial support of the EU H2020 Brightness Project, grant agreement 676548.

**Acknowledgments:** We thank our technician colleagues from CEA/Saclay for many months of collaborative work, R. de Oliveira and his team at CERN for the manufacturing of the microbulk and GEM detectors. We also acknowledge the support of LLB staff during our tests of our detector prototype at Orphée reactor as well as all our colleagues from Linköping University and the ESS detector coatings workshop who participated in the depositions of our meshes.

**Conflicts of Interest:** The authors declare no conflict of interest. The founding sponsors had no role in the design of the study; in the collection, analyses, or interpretation of data; in the writing of the manuscript, and in the decision to publish the results.

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
