# Peer review of "Large High-Efficiency Thermal Neutron Detectors Based on the Micromegas Technology"

_universe, doi:10.3390/universe4120134_

Round 1

Reviewer 1 Report

The paper is devoted to the description of various research activities connected to the realization of alternative schemes of thermal neutron detection.
Overall, the quality of the work is very good; the research is presented in a rigorous and consistent way  and the results illustrated clearly. Some of the figures need some editing, as well as the text (minor typos and/or misspells); for example, on page 2 most of the subscripts are gone, and on page 5 (line 6 of the 2nd paragraph) "responds" should read "response". Of course this list is far from being exhaustive and I am just signalling a few problems to be fixed here and there. The English is reasonable though sometimes a bit involuted, but I understand this may be  just a matter of personal taste.
I only have a point (not touched in the paper) which I find worth mentioning (and investigating): no mention is made of the gamma sensitivity of the various configurations presented. With an eye to future applications to measurements to be taken at spallation neutron sources, where a robust gamma background is always present at the location of experimental stations, an assessment of this sensitivity is mandatory. Given the generally small volume of the sensitive region of the detectors the effect should not be serious, but for the presence of a low-energy background which in turn could affect both resolution and count rate capability of the detector; with large-area detectors this could become a problem. May be a few lines should be added to the text in order to mention the problem (to be investigated in a future publication).  So far, I understand that data have been (intentionally) taken using a Cf source where neutrons are fission-generated with a low gamma-background; using AmB and AmBe sources in future measurements should allow one to investigate both effects at the same time. I have serious reasons to believe that the Authors intend to pursue this problem and their intention was to investigate only neutron detection capabilities in the first place.

Author Response

Dear Sir,

Thanks a lot for the overall positive review of the paper.

Concerning your points now:

1) I had a look at the page 2 and it seems to me that the subscripts are all there!

2) Concerning the gamma sensitivity of the proposed detection system. I have updated the paper including the following paragraph:

An important issue that was also investigated in [1] is the possible contamination of the neutron induced signal from gammas and electronic noise using a gamma source 60Co. The results showed that the expected contribution is very low and could be erased by setting an energy threshold as low as ~ 50 keV. Thus, the proposed detector, depending of the applied voltages, can be completely insensitive to gammas [21].

and I added a new reference:

[21] Fig.8 on: T. Papaevangelou et al., ESS nBLM: Beam Loss Monitors based on Fast Neutron Detection. 61st ICFA Advanced Beam Dynamics Workshop on High-Intensity and High-Brightness Hadron Beams, Jun 2018, Daejeon, South Korea. pp.THA1WE04, 2018, https://dx.doi.org/10.18429/JACoW-HB2018-THA1WE04

Best regards

Georgios

Reviewer 2 Report

It would be better to compare the advantages and disadvantages of this technology with the solid-state thermal neutron detector. The solid-state thermal neutron detector such as Boron or LiF coated microstructured Si detectors are available with efficiency greater than 30%, which requires no bias. How does this work advantageous over solid-state technology? This type of explanation provides the overall current technology for thermal neutron detection technology. Furthermore, the He-3 shortage is no more crisis.   

Author Response

Dear Sir,

Thanks a lot for the overall positive review.

Concerning your points now:

1) I added the following lines in the introduction section:

Compared with the solid state technology like boron or LiF coated microstructured Si thermal neutron detectors, Micromegas has the advantage that it is relatively easy to make large surfaces with low cost. Moreover, it is a radiation hard detector that can operate in very high particle fluxes, has fast signals (~50 ns duration) and it can be insensitive to gammas.

2) Concerning the end of He3 crisis I will add in the introduction "at that time": Due to the so-called 3He shortage crisis at that time, a coordinated initiative was launched ...

3He  price did not increase above 3k€ the litter and is now back to  1k€/liter, which is not a pb for small detector, but  is for big ones if you want high detection efficiency. It is difficult  to predict the evolution of the price in the future since the available  world 3He volume is small and the request quite changing from year to  year. However, it seems that the development  of alternative technologies for neutron detection was sufficient to  decrease the price of 3He by limiting the 3He world request...

Thanks a lot in advance

Best regards

Georgios

Reviewer 3 Report

The paper shows nice improvements on a technology (Micromegas) useful to create large area neutron detectors without utilizing 3He.

The paper is extensive and details the three main approaches followed by the experimenters to achieve a big coverage, large efficiency and scalability. 

The text is very detailed, and as such could use a couple of clearly labelled figures with schematic drawings of the assemblies being described (such as Fig 1). There are a couple of very minor errors that should be fixed about the text:

In the introduction:

"an considerable" --> "a considerable"

"using other neutron converters than" --> "using neutron converters other than"

In paragraph 4

a word is missing. One of the cited advantages is "low material". I guess it means "low material usage" or something like that.

Other than that, the text is in my opinion fine, well referenced and thorough.

The topic of the figures is my main complaint to this paper. Figures 1 and 3 are ok. Figure 2 (in particular the left side) could be improved for clarity with axes labels to give an immediate impression of the scale. The last three figures (#4 in particular) must, in my opinion, be improved significantly before publication. They are all based on screenshots (#4 even shows the scrollbar at the bottom), with very small barely visible numbers on the axes and are, all in all, confused and confusing. #4 has some weird green shadings on two of the textboxes, #5 has a double question mark (??) and an insert that is not described in the caption and #6 has some blue text on blue background which is barely visible (and blue/red on green is not clear either).

My suggestion is therefore the following:

Upgrade figure 4-5-6 (and 2 if possible) for readability.

If the authors see it fit, include a schematic drawing of the MicroBulk technology and of the GEM setup as well, with references through the text for clarity.

Fix the aforementioned typos.

Once that is done, I recommend the text for publication.

Author Response

Dear Sir,

Thanks a lot for your overall positive review.

As you could see at the attached pdf, I have corrected the typos.

The figures 4,5,6 are also improved with bigger size and more clarity

removing shaded boxes but I kept the same screenshot format.

These results are from mca but are also reproduced and verified  with an independent pulse height analysis. I can understand that its too complex for someone to understand easily what is going on but I am afraid that no more things can be done due to the limited number of pages that the submitted document should have. Anyway, more details are given in the reference [1].

Thank you very much in advance

Best regards

Georgios
